# Analytical Technique Optimization on the Detection of β-cyclocitral in *Microcystis* Species

**DOI:** 10.3390/molecules25040832

**Published:** 2020-02-14

**Authors:** Ryuji Yamashita, Beata Bober, Keisuke Kanei, Suzue Arii, Kiyomi Tsuji, Ken-ichi Harada

**Affiliations:** 1Graduate School of Environmental and Human Science, Meijo University, 150 Yagotoyama, Tempaku, Nagoya 468-8503, Japan; kiharada@meijo-u.ac.jp; 2Faculty of Pharmacy, Meijo University, 150 Yagotoyama, Tempaku. Nagoya 468-8503, Japan; beata.bober@uj.edu.pl (B.B.); 130973121@ccalumni.meijo-u.ac.jp (K.K.); takoma@kfx.biglobe.ne.jp (S.A.); 3Department of Plant Physiology and Development, Jagiellonian University, Gronostajowa 7, 30-387 Krakow, Poland; 4Kanagawa Prefectural Institute of Public Health, 1-3-1 Shimomachiya, Chigasaki, Kanagawa 253-0087, Japan; tsuji.p68c@pref.kanagawa.jp

**Keywords:** analysis, *Microcystis*, SPME, volatile organic compounds

## Abstract

β-Cyclocitral, specifically produced by *Microcystis*, is one of the volatile organic compounds (VOCs) derived from cyanobacteria and has a lytic activity. It is postulated that β-cyclocitral is a key compound for regulating the occurrence of cyanobacteria and related microorganisms in an aquatic environment. β-Cyclocitral is sensitively detected when a high density of the cells is achieved from late summer to autumn. Moreover, it is expected to be involved in changes in the species composition of cyanobacteria in a lake. Although several analysis methods for β-cyclocitral have already been reported, β-cyclocitral could be detected using only solid phase micro-extraction (SPME), whereas it could not be found at all using the solvent extraction method in a previous study. In this study, we investigated why β-cyclocitral was detected using only SPME GC/MS. Particularly, three operations in SPME, i.e., extraction temperature, sample stirring rate, and the effect of salt, were examined for the production of β-cyclocitral. Among these, heating (60 °C) was critical for the β-cyclocitral formation. Furthermore, acidification with a 1-h storage was more effective than heating when comparing the obtained amounts. The present results indicated that β-cyclocitral did not exist as the intact form in cells, because it was formed by heating or acidification of the resulting intermediates during the analysis by SPME. The obtained results would be helpful to understand the formation and role of β-cyclocitral in an aquatic environment.

## 1. Introduction

Cyanobacterial blooms are increasing worldwide [1,2] and have a negative impact on the water environment and the quality of drinking water. Cyanobacteria are able to biosynthesize and release a number of bioactive metabolites into the surrounding water. One of the characteristic groups is the group of volatile organic compounds (VOCs), such as β-cyclocitral, geosmin, and 2-methylisoborneol [3,4,5,6]. Among the VOCs, β-cyclocitral is specifically derived from *Microcystis*, which is a major genus in cyanobacterial blooms [6] and has lytic activity against cyanobacteria [7]. In addition, β-cyclocitral has been reported to be used as a repellent for *Daphnia* [8]. β-cyclocitral causes a characteristic color change in the culture broth from green to blue during the lysis process [7,9]. This phenomenon was occasionally observed under natural conditions [10] and was suggested to be caused by acid stress with 2,6,6-trimethylcyclohexene-1-carboxylic acid (β-cyclocitric acid) that was readily converted from β-cyclocitral [11,12].

Nor-carotenoids, such as β-cyclocitral and β-ionone, are known as metabolites derived from β-carotene. These compounds have been reported to be signal products formed in response to strong photo-oxidative stress in plants, and to be produced upon oxidative cleavage by CCD (carotenoid cleavage dioxygenase) enzymes. The produced nor-carotenoids are involved in the induction of a protective function against strong light, and β-cyclocitral has been specifically suggested to have a more important function [13,14]. There are several types of CCDs in a large variety of organisms, each of which specifically cleaves oxidatively β-carotene [15]. According to Cui et al. [16], there are several types of CCDs in cyanobacteria, and *Microcystis* has been reported to possess CCD7 and apo-carotenoid oxygenase (ACO). These CCDs produce β-ionone and retinal, but have not been reported to produce β-cyclocitral.

The phenomena of the blue-colored lakes have been reported, suggesting that it was induced by density stress [17,18,19]. The detected amounts of β-cyclocitral increased with the occurrence of this phenomenon [10]. However, the detection of β-cyclocitral and its oxidation product, β-cyclocitric acid, depended on the analytical method, solid phase micro-extraction (SPME) GC/MS or solvent extraction GC/MS [20]. Arii et al. reproduced the lysis phenomenon with the blue color formation by concentrating the cyanobacteria using the cylinder method [21]. A plausive mechanism was reported for this series of phenomena. Initially, density stress of the cyanobacteria occurs and lysis is caused by the production of β-cyclocitral [7]. The released β-cyclocitral is easily oxidized [12] and chlorophyll and carotenoids are degraded by the resulting acid stress, affording the blue-colored lake due to phycocyanin under favorable conditions [11]. The stressed cyanobacteria finally converge on the species of *Microcystis* [22]. Moreover, this phenomenon was reproduced by the floating concentration of cyanobacteria [21]. These phenomena are expected to be some of the survival strategies using β-cyclocitral by *Microcystis* and β-cyclocitral, and related compounds are involved in the regulation of the occurrence of cyanobacteria [23]. Therefore, the development of an analysis method of these VOCs is highly required to understand their role in cyanobacteria under natural conditions.

According to the review by Watson et al., many volatiles and nor-carotenoids can be detected using common analytical methods, such as headspace solid phase micro-extraction (HSPME), liquid–liquid extraction, or closed-loop stripping analysis coupled with gas-chromatography-mass spectrometry [6]. Jüttner reported that β-cyclocitral was produced after the cells were ruptured [3] and Zhang et al. assumed that β-cyclocitral was produced during the analysis process [9]. We also applied SPME GC/MS to the analysis of β-cyclocitral, β-cyclocitric acid, and accompanying compounds from a *Microcystis* strain. While β-cyclocitral could be certainly detected, β-cyclocitric acid could not be detected at all. However, the result was completely opposite using the typical extraction method [20].

SPME was developed by Pawlyszyn and coworkers in the early 1990s and is a solid phase extraction analysis without using solvents [24]. In this method, a fiber is exposed to an aqueous or gaseous sample until equilibrium is reached between the analyte in the sample and on the fiber. The SPME technique has been widely used in various fields, such as in the environment, food, natural products, pharmaceuticals, biology, toxicology, and forensics [25,26,27]. The parameters affecting the extraction efficiency are the nature and dimensions of the fiber coating, volatility of the analytes, extraction temperature, sample stirring rate, sample ionic strength, salting out, etc. [28]. As already mentioned, β-cyclocitral could be detected using only SPME, whereas it could not be found at all using the typical extraction method [20]. In this study, we investigated why β-cyclocitral was detected using only SPME GC/MS. In particular, three parameters, including extraction temperature, sample stirring rate, and the effect of salt, were examined for the production of β-cyclocitral.

## 2. Results

### 2.1. Analysis of VOCs Using SPME and the Modified Extraction Methods for Confirmation of the Reproducibility of the Previous Results

As mentioned in the introduction, we applied SPME GC/MS to analyze the β-cyclocitral and β-cyclocitric acid from the *Microcystis* strain NIES-298 [20]. While β-cyclocitral could be definitely detected, β-cyclocitric acid could not be detected at all. However, the result was completely opposite using the typical extraction method. Table 1 summarizes the analysis results by the two methods in which samples from 2008 to 2013 were collected from lakes and samples from 2014 and 2016 were derived from the *Microcystis* strain NIES-843. The obtained results clearly show that the previous results are completely reproduced. That is, β-cyclocitral can be detected using SPME, whereas β-cyclocitric acid can be mainly detected by the modified extraction method.

### 2.2. Factors Affecting the Formation of β-Cyclocitral during SPME

The SPME experiment was typically carried out by additional operations, including heating, salting out, and shaking [26]. The present experiment was done to recognize which operation is effective for detecting β-cyclocitral. As shown in Table 2, there are large amounts of β-cyclocitral in trials 1) to 5) which mainly included the heating operation. The highest amount was obtained in trial 3), and the average amount was estimated to be approximately 10 µg/L under the heating conditions. In contrast to these results, β-cyclocitral was not detected in trials 6) and 7), which included only salting out or shaking. This experiment indicated that heating was essential for the detection of β-cyclocitral.

### 2.3. Optimal Heating Conditions for the Detection of β-Cyclocitral by the Solvent Extraction Method

In this experiment, we investigated the heating conditions. Cultured NIES-843 was used in two different experiments at room temperature or at 60 °C. In the control at room temperature, the obtained amounts of β-cyclocitral, β-ionone, and β-cyclocitric acid were 0.44, 1.17, and 0.49 µg/L, respectively. The amounts of β-cyclocitral only increased to 8.71 µg/L by heating at 60 °C, indicating that the amount of β-cyclocitral was significantly increased by the applied heat (Figure 1).

In the next step, we tried to examine a suitable temperature and time duration for the heating. Figure 2A shows the analysis results at each heating time, and β-cyclocitral and β-ionone were increased when heated for 5 min or longer. Figure 2B shows the analysis results at each temperature. The optimized temperature for the detection was 60 °C.

### 2.4. Effect of Acidification on the Formation of β-Cyclocitral

In our extraction method, the VOCs were extracted from a cultured broth using *tert*-butyl methyl ether (MTBE) soon after acidification with 1M HCl [20]. In this experiment, we examined the effect of acidification. After the acidified samples were extracted, they were analyzed at intervals of 0.5, 1, 3, 6, and 24 h. Figure 3 shows the results of β-cyclocitral, β-cyclocitric acid, and β-ionone. The amounts of the β-cyclocitral alone increased by the acidification as follows: 15.66 µg/L, 58.60 µg/L, 59.22 µg/L, 59.93 µg/L, and 53.09 µg/L after 0.5, 1, 3, 6, and 24 h, respectively. After one hour, the amount of β-cyclocitral did not increase. These results strongly indicated that acidification with a 1-h storage time was more effective than heating when comparing the obtained amounts.

### 2.5. Comparison of the Established Methods with Typical SPME for Formation of β-Cyclocitral

We compared the non-heat SPME (A), typical SPME (B), the modified extraction with acidification (C), the modified extraction with heating (D), and the typical extraction (E) using β-cyclocitral, β-cyclocitric acid, and β-ionone (Figure 4). The obtained amounts of β-cyclocitral were 6.79 µg/L, 44.92 µg/L, 82.59 µg/L, 24.68 µg/L, and 0 µg/L, respectively. The amounts of β-ionone were 0.19 µg/L, 0.77 µg/L, 5.92 µg/L, 8.59 µg/L, and 5.18 µg/L, respectively, and the amounts of β-cyclocitric acid were 0 µg/L, 0 µg/L, 20.45 µg/L, 13.40 µg/L, and 10.13 µg/L, respectively. The results indicated that the modified extraction with acidification was quite superior to the other methods and acidification was very effective for the formation of β-cyclocitral.

## 3. Discussion

The purpose of this study was to clarify the more detailed functions of β-cyclocitral, and to verify how the difference in analytical methods contributes to the detection of β-cyclocitral. Table 1 summarizes the analysis results of VOCs by GC/MS using SPME and the typical extraction method. We confirmed that the compounds detected from the lakes and laboratories depended on the analytical conditions. That is, β-cyclocitral can be analyzed only by SPME, and β-cyclocitric acid can be analyzed only by the solvent extraction method. However, β-ionone, which is reported to be able to be produced by most cyanobacteria, can be analyzed by either method. The extraction mechanism for SPME is explained as follows: The fiber is exposed to an aqueous or gaseous sample until equilibrium is reached between the analyte in the sample and on the fiber. The analyte is then desorbed from the fiber at a gas chromatograph injector by a high temperature, and subsequently analyzed by GC/MS. In order to further shift this equilibrium, the following three operations, i.e., heating, salting out, and shaking of the liquid phase, were performed [27]. In the modified solvent extraction method, *tert*-methyl butyl ether (MTBE) was added after acidification and the extracted VOCs were analyzed using GC/MS. 

The first experiment in the present study was planned to verify which operation more effectively contributed to the formation of β-cyclocitral. As shown in Table 2, β-cyclocitral was detected in all the trials including heating, but was not detected by salt or shaking alone, indicating that heating is critical for the formation of β-cyclocitral. However, when a combination of both operations (trial 5) was used, β-cyclocitral was detected at 6.6 μg/L. In the second experiment, the heating operation at 60 °C for 20 min was added to the modified solvent extraction method. Using this method, the amount of only β-cyclocitral increased from 0.44 µg/L to 8.71 µg/L, whereas no significant difference was observed for the amounts of β-ionone and β-cyclocitric acid, as shown in Figure 1. Because heating is essential for the detection of β-cyclocitral, we examined which conditions were suitable in terms of temperature and time as the third experiment. As shown in Figure 2, 60 °C gave the best results and 5 min or more was sufficient.

In the fourth experiment, we examined whether or not acidification was effective for the formation of β-cyclocitral using the modified solvent extraction method. Figure 3 demonstrates that favorable results were obtained in which β-cyclocitral was significantly detected versus just heating. It should be noted that this method required storage for at least one hour or more. Overall, these results verified that heating or acidification was essential for the formation of β-cyclocitral. In order to further confirm these results, we compared the non-heat SPME, typical SPME, the modified extraction with acidification, the modified extraction with heating, and the typical extraction using β-cyclocitral, β-cyclocitric acid, and β-ionone (Figure 4). Reproducible results were obtained and the modified extraction with acidification provided the most sensitive results.

There were two issues to be solved in the present study. The first issue was to elucidate why β-cyclocitral could be only detected by SPME. Although β-cyclocitral is considered to be produced by related enzymes similar to CCD and ACO, no responsible enzyme has been elucidated. Harrison and Bugg described possible dioxygenase mechanisms for the cleavage reaction catalyzed by the CCDs [29]. According to their report, the oxidative cleavage reaction by CCD proceeds through two pathways, which have relatively stable intermediates (Appendix A). The intermediate-1 requires a proton for cleavage to the final product. The intermediate-2 is a dioxetane derivative, which may be labile due to the ring strain. The obtained results in this study suggested that β-cyclocitral did not exist as the intact form in cells, but it was formed by heating during the analysis by SPME. Indeed, the detection of β-cyclocitral was attained by using an instrument with a heating device in almost all the studies [3,5,9]. As shown in Figure 3 and Figure 4, acidification was superior to heating for the sensitive detection of β-cyclocitral, probably due to the fact that intermediate-1 was predominant over intermediate-2 in the cells (Appendix A). Another issue was to resolve why β-cyclocitric acid could not be detected by SPME, but could be detected by the modified extraction method. This was dependent upon the pretreatment of a sample. In the case of SPME, β-cyclocitric acid existed as the ionic form because the pH of a natural sample including cells ranged from 8 to 10. As already mentioned, the pH was adjusted to 3 with hydrochloric acid in the modified extraction method.

Based on the above discussion, the SPME technique was less effective and difficult to apply for the quantification of β-cyclocitral. Instead, we propose a practical procedure using the modified extraction for the sensitive analysis of β-cyclocitral in cells. This procedure is as follows: 2 mL of a sample is added to the screw tube and 20 µL of 1 M hydrochloric acid is added for acidification. After storing for 1 h, add 2 mL of a MTBE solution, shake using a vortex mixer, and the MTBE layer is used for analysis by the typical GC/MS.

We have recognized that β-cyclocitral is a key compound for regulating the occurrence of cyanobacteria and related microorganisms in an aquatic environment. However, no reasonable explanation has been reported concerning how β-cyclocitral is produced and degraded. As already reported, because it is insufficient using only two CCD enzymes, CCD7 and ACO, to produce β-cyclocitral intermediates, additional CCD enzymes are required [16]. In the present study, we have shown that β-cyclocitral does not exist as the intact form, and the intermediates are converted to β-cyclocitral by heating and acidic conditions. However, these conditions are unusual under natural conditions. Therefore, we have to find a mechanism of the formation for β-cyclocitral in an aquatic environment. Very recently, Chia et al. [30] reported an allelopathic interaction between *Microcystis* and *Anabaena* (*Dolichospermum*). We consider that β-cyclocitral is deeply related to this phenomenon. The obtained results from the present study would be helpful to further understand the role of β-cyclocitral for *Microcystis* and their life cycle.

## 4. Materials and Methods

### 4.1. Chemicals

The volatile compounds, β-cyclocitral and β-ionone, were obtained from Nacalai Tesque (Kyoto, Japan). Geosmin-d_3_ from Hayashi Pure Chemical Industries (Osaka, Japan) was used as the internal standard for the GC/MS analysis. *tert*-Butyl methyl ether (MTBE) and hydrochloric acid for the solvent extraction were obtained from Nacalai Tesque (Kyoto, Japan).

### 4.2. Cyanobacteria Cultures

The axenic strain *Microcystis aeruginosa* NIES-843 was obtained from the National Institute for Environmental Studies (NIES), Tsukuba, Japan. NIES-843 was cultured in 1-liter Erlenmeyer flasks containing MA medium [31] at 25 °C. The cultures were maintained under 28 µE·m^−2^·s^−1^ photo conditions until a density of 10^7^ cells/mL was reached and were later maintained under 3 µE·m^−2^·s^−1^ continuous light.

### 4.3. Analysis Procedure to Examine which Operations Affected the Formation of β-Cyclocitral during SPME

The cultured broth (1 mL) was added to a vial which was sealed with a Teflon-lined cap containing 4 g of dried NaCl. Geosmin-d_3_ solution (10 µL, 10 mg·L^−1^) was added as the internal standard. After being diluted to 10 mL with water, these vials were placed on a stirrer equipped with a heater block preheated at 60 °C. The outer needle of the SPME fiber, 50/30 µm DVB/CAR/PDMS (divinylbenzene/carboxen/polydimethyl siloxane, Supelco, Bellefonte, PA, USA), assembly was passed through the septum and the fiber extended into the head-space. In the experiment, all of the following combinations were carried out: heating (60 °C, 20 min), salting out (4 g of NaCl), and ultrasonic shaking (20 min). After 20 min, the fiber was removed and placed in the injector of a gas chromatograph (Agilent 7890B, Agilent Technologies, CA, USA) with the SPME system (CTC analytics Combi PAL autosampler, Agilent Technologies, CA, USA). The GC/MS operation was performed using a J&W Scientific column, HP-INNOWax (30 m × 0.25 mm ID × 0.25 µm film) and a mass selective detector (Agilent 5977A). Fiber desorption was performed in the splitless mode at 260 °C for 10 min. The injector temperature was 260 °C and the column temperature program was 40 °C (5 min), from 40 °C to 210 °C at 10 °C·min^−1^ and a 5 min hold at 210 °C. Helium was used as the carrier gas (1 mL·min^−1^). The detector temperature was 230 °C. Electron ionization (EI) was used for the ionization and for the selected ion monitoring (SIM) mode; *m*/*z* 152 and 137 for β-cyclocitral; *m*/*z* 177 and 135 for β-ionone; *m*/*z* 115 for geosmin-d_3_ were monitored. This experiment was carried out at *n* = 1 and the procedure was based on Fujise [20].

### 4.4. Analysis Procedure to Examine Optimal Heating Conditions for the Detection of β-cyclocitral by the Solvent Extraction Method

Cultured broth (2 mL) was added to the screw tube and heated in an electric water bath. After heating, the screw tube was cooled with tap water to room temperature. Twenty microliter of 1 M hydrochloric acid was then added for acidification. Immediately, 2 mL of MTBE solution containing geosmin-d_3_ (100 µg·L^−1^) was added. After shaking using a vortex mixer, the MTBE layer was analyzed. The GC/MS operation was performed as follows: gas chromatograph (Agilent 7890B, Agilent Technologies, CA, USA), a J&W Scientific column HP-INNOWax (30 m × 0.25 mm ID × 0.25 µm film), and a mass selective detector (Agilent 7693) were used in the splitless mode. The injector temperature was 250 °C and the column temperature program was 40 °C (1 min), from 40 °C to 240 °C at 20 °C·min^−1^ and a 10-min hold at 240 °C. Helium was used as the carrier gas (1 mL·min^−1^). The detector temperature was 230 °C. EI was used for the ionization and for the SIM mode; *m*/*z* 152 and 137 for β-cyclocitral; *m*/*z* 177 and 135 for β-ionone; *m*/*z* 153 and 107 for β-cyclocitric acid; *m*/*z* 115 for geosmin-d_3_ were monitored. The experimental procedure was based on Fujise [20]. For the standard heating condition, 60 °C for 20 min was applied at *n* = 7. Furthermore, various other heating conditions for 0, 5, 10 15, 20, 25, 30, 40, 50, 60, 90, and 120 min at 60 °C in the durations as well as at 20, 30, 40, 50, 60, 70, and 80 °C in temperature for 20 min, respectively, were also examined (*n* = 3). The average and 95% confidence intervals were plotted on a graph.

### 4.5. Analysis Procedure to Examine Effect of Storage Time after Acidification on the Formation of β-Cyclocitral

The cultured broth (2 mL) and 20 µL of 1 M hydrochloric acid were added to the screw tube for acidification. The tubes were stored for 0, 0.5, 1, 3, 6, 24 h before extracting with 2 mL of the MTBE solution containing geosmin-d_3_. After shaking using the vortex mixer, the MTBE layer was analyzed. The GC/MS operation was already described. This experiment was carried out at *n* = 3 and the average and 95% confidence intervals were plotted on a graph.

### 4.6. Comparison of the Established Methods with the Typical SPME for Formation of β-Cyclocitral

The following five methods were examined for the production of β-cyclocitral: (A) SPME at room temperature (non-heat SPME), (B) typical SPME for control (heating condition: 60 °C for 20 min), (C) solvent extraction after acidification with storage for 3 h, (D) solvent extraction after heating at 60 °C for 20 min, and (E) solvent extraction at room temperature. Each method was repeated three times, and the average and 95% confidence intervals were plotted on a graph.

## 5. Conclusions

In the present study, we investigated why β-cyclocitral was detected by only SPME GC/MS and revealed that the SPME operating conditions strongly influenced the formation of β-cyclocitral. β-Cyclocitral was formed by heating at 60 °C for 5 min or more or by acidification for 1 h, and more effectively under acidic conditions. It should be emphasized that β-cyclocitral does not exist as the intact form in cells, because it is formed during the analytical process. β-Cyclocitral was retained as two plausible intermediates from the CCD reaction and is formed by heating, acidification, or similar conditions in a water environment. In this study, we also proposed a practical analysis method for β-cyclocitral and related compounds in cells, which has the characteristic feature that it is possible to simultaneously analyze β-cyclocitral and β-cyclocitric acid by a simple operation, including acidification.

## Figures and Tables

**Figure 1 molecules-25-00832-f001:**
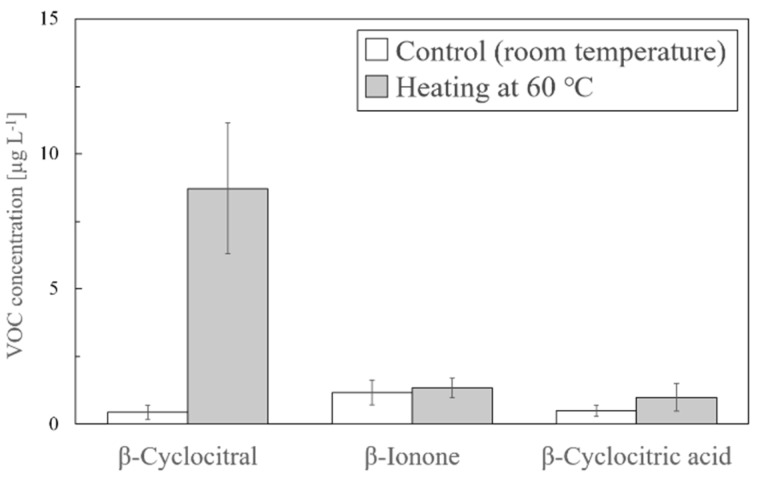
Analysis of three VOCs by the solvent extraction with heating (*n* = 7).

**Figure 2 molecules-25-00832-f002:**
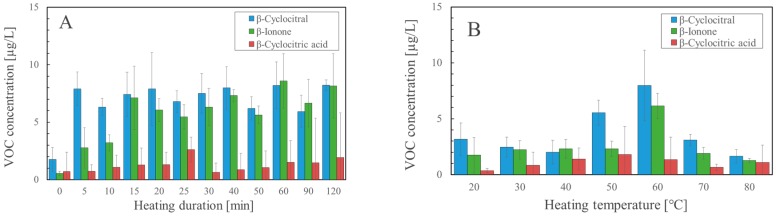
Optimization of heating conditions for detection of β-cyclocitral by the modified solvent extraction (*n* = 3). (**A**) Heating duration, (**B**) Heating temperature.

**Figure 3 molecules-25-00832-f003:**
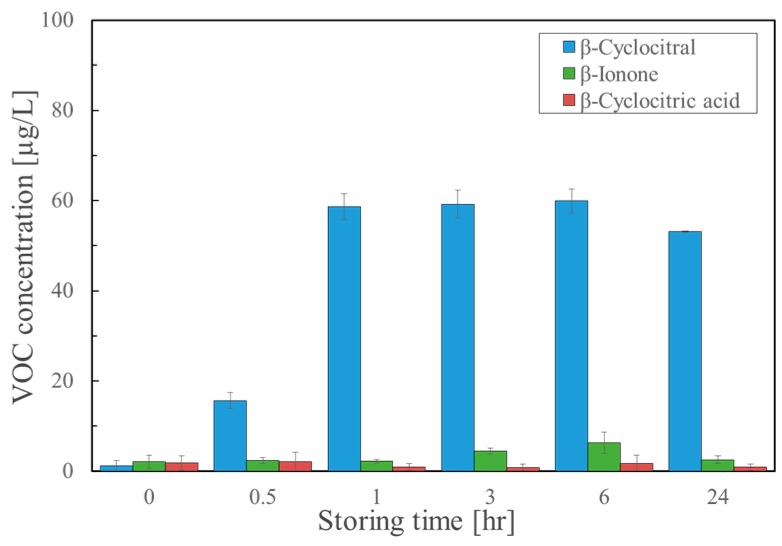
Analysis results of β-cyclocitral, β-ionone, and β-cyclocitric acid after acidification followed by storing for 0.5, 1, 3, 6, and 24 h (*n* = 3).

**Figure 4 molecules-25-00832-f004:**
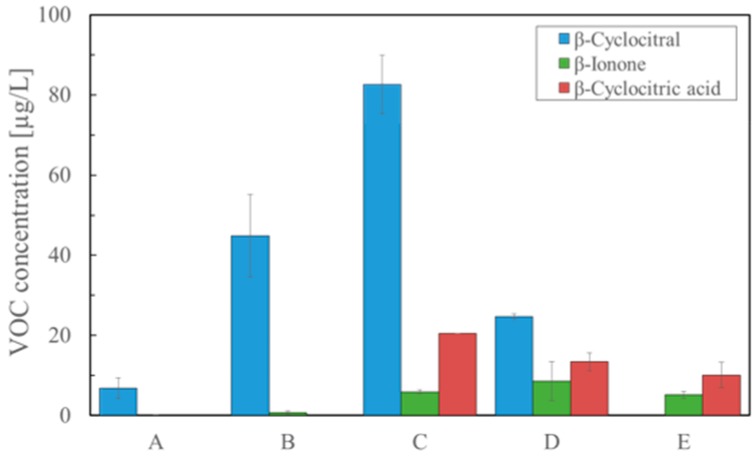
Analysis results β-cyclocitral, β-ionone, and β-cyclocitric acid using the following operations (*n* = 3). A: Non-heat SPME, B: SPME, C: Solvent extraction after acidification, D: Solvent extraction with heating, E: Solvent extraction.

**Table 1 molecules-25-00832-t001:** Analysis results of volatile organic compounds (VOCs) using solid phase micro-extraction (SPME) and the typical solvent extraction methods from 2008 to 2016.

	2008		2008		2010		2010		2013		2014		2016	
	Total VOC	Filtrate VOC	Total VOC	Filtrate VOC	Total VOC	Filtrate VOC	Total VOC	Filtrate VOC	Total VOC	Filtrate VOC	Total VOC	Filtrate VOC	Total VOC	Filtrate VOC
Method	SPME		SPME		SPME		SPME		S.E.	S.E.	SPME	S.E.	SPME	S.E.
Cell number [cells/mL]	6.2 × 10^4^		3.1 × 10^4^		4.2 × 10^7^		7.6 × 10^7^		2.0 × 10^5^	8.5 × 10^5^	2.0 × 10^7^		8.4 × 10^6^	
pH	6.7		5.4		5.9		9.9		8.7	6.2	10.0		10.0	
β-Cyclocitral [µg/L]	50	0.9	100	55	1400	3.2	52	1.2	ND	ND	136	ND	93	ND
β-Cyclocitric acid [µg/L]	-	-	-	-	-	-	-	-	52	120	ND	36	-	20
β-Ionone [µg/L]	9.6	7.6	98	15	300	1.6	26	0.5	120	150	15.7	-	22	ND

S.E.: Typical solvent extraction, -: Not determined, ND: not detected.

**Table 2 molecules-25-00832-t002:** Analysis results of β-cyclocitral from *Microcystis aeruginosa* NIES-843 using GC/MS with SPME by the appropriate combination of heating, salting out, and shaking (*n* = 1).

Trial No.	Heat	Salt	Shake	β-Cyclocitral [µg/L]
1)	+	+	+	9.7
2)	+	+	-	9.3
3)	+	-	+	14.3
4)	+	-	-	12.3
5)	-	+	+	6.6
6)	-	+	-	0
7)	-	-	+	0

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
