# Peer review of "Analytical Technique Optimization on the Detection of β-cyclocitral in Microcystis Species"

_molecules, 2020, doi:10.3390/molecules25040832_

Round 1

Reviewer 1 Report

The paper by Yamashita et al. describes the determination of β-cyclocitral using SPME GC-MS. Different SPME operation conditions were investigated. The results obtained by the authors could be helpful to understand the formation of  β-cyclocitral in aquatic environments.

I suggest the following explanations/amendments

some english grammar corrections table 1 is not clear table 2: please add the standard deviations for each concentration Page 5, L135-156: this part of the text would be more appropriate in the Introduction

I recommend a minor review before to consider the manuscript for publication.

Author Response

Response to Reviewer 1 Comments

Thank you for providing these comments.

Point 1: Some English grammar corrections table 1 is not clear

We have revised the text in Table 1 Lines 100-102 in the revised manuscript:

・Units are listed in the table.

・”usual solvent extraction” to “typical solvent extraction”

・”Usual” to “S.E.” and add the following legends, “S.E.: Typical solvent extraction”

Point 2: Table 2: please add the standard deviations for each concentration

This experiment was performed by GC/MS (Agilent 6890), but this was unfortunately broken and it was impossible to estimate standard deviations. Subsequently, we have changed to GC/MS (Agilent 7890) shown in Materials and Methods and we can have reproducible results by both machines.

Point 3: Page 5, L135-156: this part of the text would be more appropriate in the Introduction

According to the suggestion, this part has been moved to the Introduction Lines 44-68 in the revised version.

Reviewer 2 Report

I have carefully read the manuscript. The authors have performed an excellent job regarding the conducted study. Manuscript is well written and organized, while the results are very significant for the scientific community. Therefore, I recommend an acceptance of this manuscript for publication in Molecules after the minor revision regarding the English language and grammar.

Author Response

Response to Reviewer 2 Comments

Thank you for your positive feedback on our paper.

We will ask English correction by MDPI after the acceptance.

Reviewer 3 Report

Comments on manuscript molecules-678578

Page 1

The title should be changed to “Optimization of analytical technique on the detection of β-cyclocitral in Microcytis species”. This would better reflect the objectives of the manuscript.

Line 34: change to “…able to biosynthesize and release a number of bioactive metabolites into the surrounding water.”

Line 37: change to “Among the VOCs, β-cyclocitral is specifically derived from Microcystis, which is a major genus in cyanobacterial blooms [6] and has lytic activity against cyanobacteria.”

Page 2

Line 49: change to “…many of volatiles and nor-carotenoids can be…”

Line 67: change to “In particular, three parameters, including extraction temperature, sample stirring rate and effect of salt, were examined for the production of β-cyclocitral.”

Line 66: The authors stated that their study investigated “why and how” β-cyclocitral was detected using only SPME GC/MS. I think the main aim of their study is to optimize the procedure for the extraction of the molecule. I don’t think the study explore the “why and how” the molecule was detected by SPME GC/MS. It is more of a hypothesis on why β-cyclocitral could only be detected by SPME as outlined on page 6, lines 189 – 201.

Line 71: is the “…previous results” in the subheading referring to “published results”?

Line 83: Instead of “Operations”, it should be “Factors affecting the…”

Page 3

Lines 86 – 91: The results reported in this paragraph are based on n=1. Would it be possible to increase the number of replicates?

Figure 1: The legend should be relabeled as “Control @ RT” and “Heating at 60oC”

I am not sure if Figure 1 is needed since the authors proceeded to carry out the experiments based on heating duration and heating temperature in Figure 2. Also, what was the heating temperature for Figure 2A?

Page 4

Lines 113 – 115: Based on Figure 3, the amount of β-cyclocitral increased after storing for 1h. Subsequent storage time did not significantly increase the amount of β-cyclocitral. Authors can reword this sentence to reflect the data shown in Figure 3.

Page 5

Figure 4: The figure showed the concentrations of 3 compounds and not 4 as mentioned in the caption of the figure.

Lines 163 – 165: These sentences describing the nature of silica fiber can be omitted.

Author Response

Response to Reviewer 3 Comments

Thank you for the thoughtful and constructive feedback you provided regarding our manuscript.

Page 1

Point 1: The title should be changed to “Optimization of analytical technique on the detection of β-cyclocitral in Microcytis species”. This would better reflect the objectives of the manuscript.

We would like to appreciate this advice, therefore the title has been changed to “Optimization of analytical technique on the detection of β-cyclocitral in Microcytis species” (Lines 2-3 in the revised manuscript).

Point 2: Line 34: change to “…able to biosynthesize and release a number of bioactive metabolites into the surrounding water.”

According to the comment, this sentence has been changed (Lines 34-35 in the revised manuscript)

Point 3: Line 37: change to “Among the VOCs, β-cyclocitral is specifically derived from Microcystis, which is a major genus in cyanobacterial blooms [6] and has lytic activity against cyanobacteria.”

According to the comment, this sentence has been changed (Lines 37-38 in the revised manuscript).

Page 2

Point 4: Line 49: change to “…many of volatiles and nor-carotenoids can be…”

According to the comment, this sentence has been changed (Line 69 in the revised manuscript).

Point 5: Line 67: change to “In particular, three parameters, including extraction temperature, sample stirring rate and effect of salt, were examined for the production of β-cyclocitral.”

According to the comment, this sentence has been changed (Lines 87-88 in the revised manuscript).

Point 6: Line 66: The authors stated that their study investigated “why and how” β-cyclocitral was detected using only SPME GC/MS. I think the main aim of their study is to optimize the procedure for the extraction of the molecule. I don’t think the study explore the “why and how” the molecule was detected by SPME GC/MS. It is more of a hypothesis on why β-cyclocitral could only be detected by SPME as outlined on page 6, lines 189 – 201.

We agree this suggestion and have changed “why and how” to “why” accordingly (Lines 22, 86 and 285 in the revised manuscript).

Point 7: Line 71: is the “…previous results” in the subheading referring to “published results”?

Of the results shown in Table 1, the data in 2008 and 2010 have been already published [21], but the remaining data are unpublished results.

Point 8: Line 83: Instead of “Operations”, it should be “Factors affecting the…”

According to the comment, this subtitle has been changed (Line 103 in the revised manuscript).

Page 3

Point 9: Lines 86 – 91: The results reported in this paragraph are based on n=1. Would it be possible to increase the number of replicates?

This experiment was performed by GC/MS (Agilent 6890), but this was unfortunately broken and it was impossible to increase the number of replicates. Subsequently, we have changed to GC/MS (Agilent 7890) shown in Materials and Methods. It is confirmed that we can be have reproducible results by both machines.

Point 10: Figure 1: The legend should be relabeled as “Control @ RT” and “Heating at 60oC”

According to the comment, the legend of Figure 1 has been changed (Line 125 in the revised manuscript).

Point 11: I am not sure if Figure 1 is needed since the authors proceeded to carry out the experiments based on heating duration and heating temperature in Figure 2. Also, what was the heating temperature for Figure 2A?

Based on the results in Table 2 we planned to analyze the desired VOC under temporary heating conditions set at 60°C that was used for usual SPME. This experiment was very important and repeated several times for detection of β-cyclocitral. Figure 2 shows an attempt to optimize the heating conditions. The heating temperature of Figure 2 (A) is 60 °C.

Page 4

Point 12: Lines 113 – 115: Based on Figure 3, the amount of β-cyclocitral increased after storing for 1h. Subsequent storage time did not significantly increase the amount of β-cyclocitral. Authors can reword this sentence to reflect the data shown in Figure 3.

According to the comment, the following sentence has been added as “After one hour, the amount of β-cyclocitral did not increase.” (Lines 135-136 in the revised manuscript).

Page 5

Point 13: Figure 4: The figure showed the concentrations of 3 compounds and not 4 as mentioned in the caption of the figure.

Certainly, β-cyclocitral has been repeatedly written. Therefore, the caption has been changed to “Analysis results β-cyclocitral, β-ionone and β-cyclocitric acid using the following operations (n = 3)" (Line 151 in the revised manuscript).

Point 14: Lines 163 – 165: These sentences describing the nature of silica fiber can be omitted.

According to the comment, the sentence “A silica fiber is coated with a stationary phase.” has been removed (Lines 162 in the revised manuscript).